# Quantum Matter Overview

**Melanie Swan** [1,*]**, Renato P. dos Santos** [2] **and Frank Witte** [3]

[1] Computer Science, University College London, London WC1E 6BT, UK
[2] Physics, Lutheran University of Brazil, Canoas 92425-900, Brazil; renatopsantos@ulbra.edu.br
[3] Economics, University College London, London WC1E 6BT, UK; f.witte@ucl.ac.uk
[*] Correspondence: melanie@blockchainstudies.org; Tel.: +44-20-7679-2000

**Abstract:** Quantum matter (novel phases of matter at zero temperature with exotic properties) is a growing field with applications in its own domain, and in providing foundational support to quantum sciences fields more generally. The ability to characterize and manipulate matter at the smallest scales continues to advance in fundamental ways. This review provides a plain-language, non-technical description of contemporary activity in quantum matter for a general science audience, and an example of these methods applied to quantum neuroscience. Quantum matter is the study of topologically governed phases of matter at absolute zero temperature that exhibit new kinds of emergent order and exotic properties related to topology and symmetry, entanglement, and electronic charge and magnetism, which may be orchestrated to create new classes of materials and computational devices (including in the areas of spintronics, valleytronics, and quantum computing). The paper is organized to discuss recent developments in quantum matter on the topics of short-range topologically protected materials (namely, topological semimetals), long-range entangled materials (quantum spin liquids and fractional quantum Hall states), and codes for characterizing and controlling quantum systems. A key finding is that a shift in the conceptualization of the field of quantum matter may be underway to expand the core focus on short-range topologically protected materials to also include geometry-based approaches and long-range entanglement as additionally important tools for the understanding, characterization, and manipulation of topological materials.

**Keywords:** quantum matter; topological materials; topological insulators; topological semimetals; quantum spin liquids; fractional quantum Hall effects; quasiparticles; anyons; toric code

## 1. Introduction

### 1.1. Quantum Matter

Quantum matter (topological materials, topological matter phases, quantum materials with topological properties) refers to novel phases of matter that arise at absolute zero temperature (0 kelvin) with emergent order and exotic properties. Traditionally, symmetry (Landau symmetry breaking) was sufficient for characterizing materials, however, quantum matter requires a topological description, and hence the name topological materials, or quantum materials with topological order. Topology and symmetry (symmetry breaking), and entanglement, are the main tools for describing, creating, and manipulating quantum matter phases. Other salient attributes of quantum matter systems may include the emergence of quasiparticles (collective excitations) with anyonic exchange statistics (identical particles swapping places by changing their wavefunction in a process called 'braiding'), gauge theory, quantum phase transitions, and low-energy effective theories of topologically ordered states. Some examples of quantum matter phases include topological insulators, topological semimetals, fractional quantum Hall states, quantum spin liquids, and strongly correlated quantum liquid states. Quantum matter provides rigorous ways for treating many-body systems whose collective behavior triggers quasiparticles (excitations) and many-body localization (states in which collective interactions cause quantum particles to be localized and maintained in an out-of-equilibrium state).

One way of categorizing the quantum matter field is by research efforts that, respectively, treat the short-range protected properties and long-range entangled properties of topological materials (Table 1). Short-range protected topological order refers to quantum matter phases that are primarily characterized as having a variety of symmetry protections in the short-range, but whose entanglement properties are trivial. Long-range entangled systems, on the other hand, centrally feature entanglement as a non-local order parameter for describing interactions and correlations in many-body systems [1]. The paper is structured in sections to discuss quantum matter building blocks, scientific efforts that focus on the short-range protected and long-range entangled properties of quantum systems, codes (mathematical encoding models to measure, confirm, and correct quantum matter systems), and an example of these methods applied to quantum neuroscience.

**Table 1.** Quantum Matter Building Blocks and Short-range and Long-range Matter Phases.

| Building Blocks | Short-Range Protection | Long-Range Entanglement |
|---|---|---|
| Symmetry and topology | Topological insulators | Quantum spin liquids |
| Anyons/quasiparticles | Topological superconductors | Quantum Hall states |
| Hyperbolic space | Topological semimetals | Entanglement entropy |

The Relation of Quantum Matter and Quantum Information Science

Two of the cornerstone fields in the foundations of quantum science are quantum matter and quantum information science. Quantum matter works closely with physical materials, attempting to characterize and exploit the properties of existing and novel matter phases. Quantum information science establishes information-theoretic formulations of physics problems, for example, evaluating the information content in quantum states with entropy calculations. Some of the main topics in the research agendas for the two fields are set forth in Figure 1, together with their overlap in the use of codes and available experimental platforms for quantum computation, quantum machine learning, quantum simulation, and quantum nanoscience material fabrication. The information-theoretic formulation of codes (quantum system encodings) provides a solvable model of topological phases, for example, in toric code models, a spin-1/2 system on a square lattice with stabilizer operators whose boundary conditions are in the shape of a torus [2].

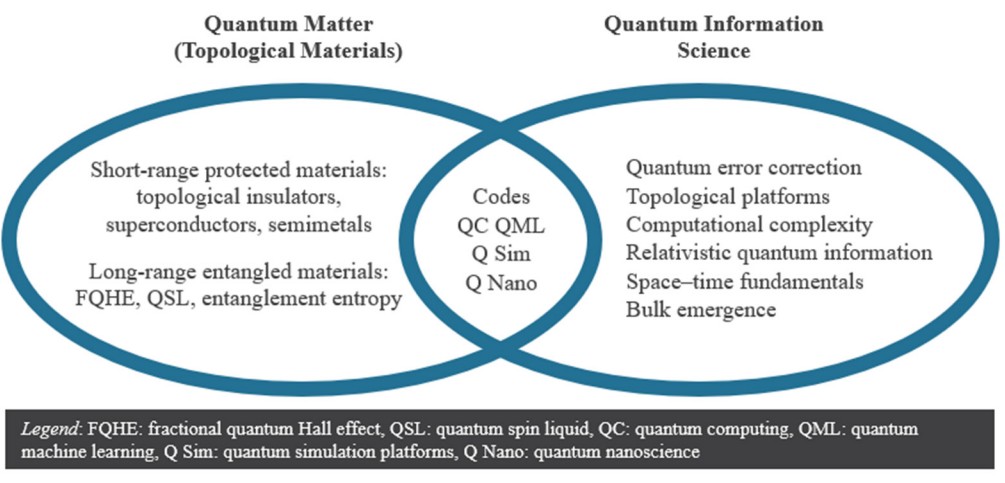

**Figure 1.** Research Topics in Quantum Matter and Quantum Information Science.

*1.2. Symmetry and Topology*

The main ways of characterizing and manipulating quantum matter are with entanglement, symmetry, and topology. Entanglement is the quantum property of correlated physical attributes among particles (position, momentum, spin, polarization). Symmetry refers to features of particles and spacetime that are unchanged under some transformation,

seen as the property of a system looking the same from different points of view (e.g., a face, a cube, or the laws of physics) and its partner, symmetry breaking (phase transition). Topology is the property of geometric form being preserved under deformation (e.g., bending, stretching, twisting, and crumpling, but not cutting or gluing). Physical systems (such as the universe and quantum mechanical systems) may have global symmetric and topological properties that remain invariant across system scales.

Symmetry typically describes the matter phases found in classical materials, such as water progressing through temperature-based transitions between its forms as ice, liquid, and vapor. In classical materials, the familiar matter phases (liquid, gas, solid) are described by the different symmetry-based ways that their constituent atoms can be organized (atoms are randomly distributed with continuous translation symmetry in a liquid, for example, and structured into a regular array (lattice) with discrete translation symmetry in a crystal). When matter undergoes a phase transition (e.g., from a liquid to a gas), the symmetry is broken and the atoms reorganize according to a different paradigm. These kinds of matter phase transitions can be described by Landau symmetry breaking [3]. Quantum matter phases, however, defy Landau symmetry breaking and require topological-based explanations of individual and group behavior.

### 1.2.1. Quasiparticles (Fermions) and Collective Excitations (Bosons)

Quantum matter opens a new era in the study of many-body systems through the analysis of emergent collective behavior known as quasiparticles (fermions) or collective excitations (bosons). Quasiparticles and collective excitations are the quantum analogue of geometric shapes made by group behavior, for example, in the macroscale context in formations such as schools of fish and flocks of birds. Quasiparticles and collective excitations emerge from collective behavior in topological phases, and may serve as artifacts or constructs for better understanding of the inner workings of these phases. Notable examples include electron holes (positively charged quasiparticles denoting the lack of an electron in a state in the valence band of a semiconductor), excitons (an electron and an electron hole bound together), phonons (vibration of atoms in a rigid crystal structure), plasmons (plasma oscillations in which electrons simultaneously oscillate with respect to ions), and magnons (a perfect alignment of magnetic moments or collective spin wave in a ferromagnet). In practical use, quasiparticles are used to simplify the calculation of the overall excitation in many-body systems, obtain the ground state and excited states of the system, and determine the bulk properties of low-energy systems.

### 1.2.2. Anyons: Third Kind of Particle

Anyons are a third kind of particle that behave like "any" generic particle, with intermediate and less-restricted values than those of the two main kinds of particles, fermions and bosons. Anyons were proposed in 1982 [4] and detected in 2020 in experimental setups involving an accelerator [5] and an interferometer [6]. The former project directed an electron gas through a small particle collider to realize fractional electric charges (the fractional Hall quantum effect, a physical system predicted to host anyon behavior), and the latter used an etched chip that screened out interactions interfering with the anyon behavior. Although anyons cannot exist as fundamental particles in nature, certain kinds of two-dimensional condensed matter systems are predicted to host exotic quasiparticles which obey anyonic statistics in which two identical anyons may swap places, changing their wavefunction in a process called 'braiding'.

Fermions and bosons are the two basic kinds of particles. Fermions are matter particles (electrons, protons, neutrons, quarks), with half-odd integer (1/2, 3/2, etc.) spin and do not like to clump together, keeping themselves spread out, for example, in electron shells in atoms (per the Pauli exclusion principle). Bosons are force particles (photons, gluons, gravitons) with integer spin, and like to clump together, for example, to form a beam of light. Whereas fermions and bosons are foundational particles, anyons arise through the

behavior of other particles, and exhibit statistical signatures in between those of bosonic bunching and fermionic exclusion.

*1.3. Hyperbolic Space (Negative Curvature)*

As Landau symmetry models break down as a model for describing topological materials, so too Euclidean space-based models such as the Bloch theorem prove inadequate. There are various forms of space. For example, planes in two-dimensional geometries may exhibit one of three forms based on their curvature: elliptic (positive curvature), Euclidean (zero curvature), or hyperbolic (negative curvature). Hyperbolic space is of interest for modeling quantum mechanical systems as more degrees of freedom can be incorporated that more realistically correspond to the complex values of wavefunctions. In negatively curved space, the angles of geometric shapes (triangles, squares) sum to less than those of their Euclidean counterparts. Whereas only four squares can connect at the vertex of a square lattice in Euclidean space, it is possible for more than four squares to connect at vertices in a hyperbolic lattice [7]. The connectivity permits a greater density of polygons at vertices, which means that more degrees of freedom can be modeled and exploited, as has been demonstrated in quantum error-correction schemes [8]. The AdS/CFT (Anti-de Sitter space/Conformal Field Theory) correspondence model is one of the first uses of hyperbolic geometry in condensed matter systems as Anti-de Sitter space is a form of hyperbolic space. Visually, hyperbolic space is depicted in Escher's *Circle Limits* works which have a circle with images of bats or other figures shrinking in a fractal-like manner as they proceed from the center to the surface.

1.3.1. Hyperbolic Bloch Theorem and Hyperbolic Band Theory

Given the benefits of hyperbolic space for more accurate wavefunction modeling, various research programs are underway to establish tools in this area. One advance is quantum simulators in the form of circuit quantum electrodynamics (cQED) as an experimental platform for modeling and synthesizing quantum matter phases in hyperbolic space [9]. Hyperbolic lattices are a lattice instantiation in hyperbolic space for the coherent propagation of wave-like excitations, essentially constituting a new form of synthetic quantum matter in which particles effectively hop on a discrete tessellation of two-dimensional hyperbolic space [10].

A foundational research effort generalizes the Bloch theorem to hyperbolic space with algebraic geometry, which entails developing a hyperbolic version of the energy band theory [11]. The Bloch theorem has been a cornerstone formulation in quantum mechanics as it casts the generally intractable Schrödinger wave equation as a solvable model of a wave propagating in a periodic (regular) system. The conventional energy band theory is also based on periodicity, computing the energy tiers of a system from the wavefunction for an electron in a periodic lattice of the material's atoms or molecules (measuring periodicity in the Brillouin zone, a geometric zone inside a crystal or lattice). Although the Bloch theorem is well defined for regular crystal structures which correspond to Euclidean geometry, it does not incorporate the full range of lattice degrees of freedom in wave phenomena, which are more accurately modeled with hyperbolic space.

The hyperbolic Bloch theorem is generalized in the hyperbolic band theory, and demonstrated by describing an example of aperiodic Hamiltonians with hyperbolic tiling symmetry. A hyperbolic version of the Brillouin zone is defined that is topologically equivalent to a higher-dimensional compact torus. The resulting hyperbolic band structure is given as a set of functions on the higher-dimensional Brillouin zone and can be computed exactly. The work is extended by giving proofs for Bloch theorems of hyperbolic lattices, and by clarifying that the hyperbolic Bloch theorem is generally non-abelian and may involve infinitely many Brillouin zones for a single lattice [10].

### 1.3.2. Magnetics

The hyperbolic band theory is likewise extended to the magnetic context, by applying an external magnetic field to a hyperbolic lattice. One team studied a hyperbolic lattice (Riemann surface) in the presence of a magnetic field to identify the energy spectrum of magnetic hyperbolic Bloch states [12]. The resulting states form Dirac cones on a coordinate neighborhood (showing the first explicit example of a massless Dirac state on a higher genus surface). The work is executed with the experimental circuit quantum electrodynamics platform. Further, algebras to support the magnetic hyperbolic band theory have been developed by proposing the magnetic Fuchsian group (a hyperbolic Riemannian surface formulation) and the magnetic hyperbolic Bloch state [13]. Another team realized a topological hyperbolic lattice under a uniform magnetic field to lead to the hyperbolic version of the quantum spin Hall effect [7]. A Euclidean photonic platform was constructed to inherit the topological band properties of a hyperbolic lattice with a magnetic field applied. The work is one of the first examples of a non-Euclidean counterpart to the quantum spin Hall effect. In the hyperbolic lattices, a high density of edge states indicated topological protection, which could be useful in photonic device design.

### 1.3.3. Condensates

The adaptation of condensed matter models to hyperbolic space continues in other domains as well, for example condensates. One project considered the Bose gas in hyperbolic space, finding that interacting Bose gases on two-dimensional and three-dimensional hyperbolic manifolds indicate Bose–Einstein condensation in the infinite-volume limit [14]. Other projects have taken an AdS/BEC approach by interpreting Bose–Einstein condensates in the AdS/CFT correspondence model of hyperbolic space. One team proposes an explanation for dark matter as a gluonic Bose–Einstein condensate in Anti-de Sitter space–time [15]. Another team studied strongly coupled Bose–Einstein condensates with a holographic model to explain disordered turbulence in the process of fractional vortex generation [16].

Condensates have been of interest since the realization of the Bose–Einstein condensate in 1995, as a quantum state of matter formed by cooling a low-density bosonic gas to near zero temperatures such that quantum mechanical effects (wavefunction interference) become visible [17]. The benefit of the Bose–Einstein condensate is amplifying microscopic quantum effects to visible macroscopic scales. Researchers are now able to condense more complex particles in the fermionic domain such as lanthanides [18], reactive metals implicated in high-temperature superconducting [19], and also in the bosonic domain, as quantum droplets comprised of bose–bose mixtures and dipolar quantum gases (supersolids) [20]. These kinds of condensate methods are used to solve localized wavefunctions with complex topological structures, and could be deployed more exploratively, for example, to study non-von Neumann architectures for information processing tasks, including to study associative memory in the brain [21].

Under the general rubric of condensates, matter phases stretch along a stratification that includes solitons (solitary wavepackets), droplets, and supersolids (spatially ordered material with superfluid properties). Condensates are used to define novel quantum matter phases resulting from emergent properties in the interactions of bosonic [22] and fermionic [18] constituent particles, for example, quantum droplets and pre-droplet solitons. A quantum droplet is a quantum system comprised of bosonic or fermionic constituents whose particle interactions cause an emergent phase of matter. Quantum droplets have interesting properties such as self-bound features, collective excitations, soliton-to-droplet transition, droplet–droplet collision, and supersolid states. One topological matter project articulates chiral edge solitons, a formulation in which chirality can be induced by adding a nonlinear term to the model of two chiral edge states with opposite chiralities [23].

## 2. Short-Range Protected Topological Materials

### 2.1. Introduction to Short-Range Protected Topological Order

A central research area in quantum matter is topological materials with short-range protected properties, studied through energy band topology, symmetry, transport, and bulk–boundary relationships. Per conventional energy band theory (the range of allowed/prohibited energy levels in a system (bands/band gaps)), solids may be classified as insulators, semiconductors, semimetals, and metals. Insulators and semiconductors are characterized by a band gap between the filled valence band and the empty conduction band. Semimetals, on the other hand, have a small region of overlap between the bottom of the conduction band and the top of the valence band. Short-range protected topological materials are thus divided into two main classes related to band gaps (topological insulators and superconductors) and overlapping bands (topological semimetals) [24].

Topological insulators and topological superconductors have gapped energy tiers in the bulk and gapless conducting modes on the surface. The surface excitations are triggered by electrons in topological insulators and by quasiparticles (coherent superpositions of electron and hole excitations) in topological superconductors [25,26]. The foundational topological material is topological insulators, which are materials with a conducting surface and an insulating interior [27]. For gapped phases, the band topology can be well-described using topological invariants in terms of symmetries [28]. As the conduction and valence bands cross each other in the Brillouin zone (geometric zone in a lattice or crystal), the system enters a semimetal phase. The band crossing gives rise to nontrivial topology that may occur at discrete points (Weyl or Dirac semimetals) or along closed loops (nodal-line semimetals).

The various forms of solids are used in device design to produce configurations in which electric currents can be controlled and flow with the lowest possible energy dissipation. Applications of topological insulators, superconductors, and semimetals include spintronics (using electron spins for information processing), valleytronics (using band structure valleys for information processing), and quantum computing. As an increasing number of topological materials continue to be discovered, classificatory systems are an indispensable tool. The surface of materials is protected by various topologically invariant properties related to the bulk. In particular, the bulk–boundary relationship is one in which bulk wavefunctions lead to surface states which can be categorized by symmetry, including both spatial (reflection, lattice) and nonspatial (time-reversal) symmetries. One project presents a comprehensive symmetry-based classificatory schema centering on five kinds of symmetry in non-interacting fermionic systems [25]. The rubric addresses time-reversal symmetry (an anti-unitary operator acting on the fermion creation and annihilation operators), particle-hole symmetry (a unitary transformation mixes fermion creation and annihilation operators), chiral symmetry (a combination of time-reversal and particle-hole symmetry), symmetry in more exotic systems (using Nambu spinors instead of complex fermion operators), and non-unitary symmetries. The classification system treats non-interacting (mean-field Hamiltonian) systems, and thus future classifications might be expanded to include interacting systems.

### 2.2. Topological Semimetals

Topological semimetals are useful as the overlap region between the conduction band and the valence band and can be manipulated to produce tunable surface states (manipulated with computable integrals). Properties can be adjusted by varying the thickness of the materials, introducing defects (impurity doping), and managing state degeneracy.

In the basic formulation of topological semimetals, the energy bands cross in a (zero-dimensional) point format or a (one-dimensional) line or ring format. Hence, the basic forms of topological semimetals are point-based Weyl and Dirac semimetals, and nodal-line semimetals. Bulk wavefunctions lead to surface energy states; for example, bulk Weyl points may result in surface Fermi arcs, and bulk nodal lines may trigger a flat surface band or drumhead state on the boundary.

### 2.2.1. Weyl and Dirac Topological Semimetals

The most straightforward realization of topological semimetals is two energy bands that cross at a single node (Weyl node) [29]. Weyl and Dirac semimetals exhibit two-fold and four-fold degenerate Fermi points, respectively. Whereas Weyl points can occur in the absence of any symmetry besides translation, Dirac points are only topologically stable in the presence of time-reversal symmetry together with a crystal lattice symmetry (such as rotation or reflection symmetry) [30]. Weyl and Dirac semimetals both exhibit arc-like surface states. In Weyl semimetals, one-dimensional surface Fermi arcs are direct topological consequences of the bulk three-dimensional Weyl points [31]. In Dirac semimetals, however, the surface Fermi arcs are not directly related to the bulk Dirac points and a more complicated topological bulk–boundary correspondence for Dirac semimetals is indicated.

### 2.2.2. Nodal-Line Topological Semimetals

Nodal-line topological semimetals are materials with energy band-touching manifolds at one-dimensional nodal lines or rings in the bulk. Such nodal lines in the bulk lead to different boundary behavior such as flat surface bands and drumhead states. One research project studied the drumhead surface state in a particular material ($Ca_3P_2$) to elaborate on how the bulk–boundary relationship gives rise to the topological protection of line nodes [30]. Various forms of symmetry are implicated including reflection, time-reversal symmetry, SU(2) spin-rotation (special unitary group), and inversion symmetry. A related projects extends the Dirac semimetal formulation in Dirac nodal-line semimetals to investigate one-dimensional Dirac nodal rings that are protected by a combination of inversion and time-reversal symmetry [32]. A small break in the inversion symmetry allows a Hall-like current to be created in which carriers at opposite sides of the Dirac nodal ring flow to opposite surfaces when an electric field is applied. The result is the formulation of a Berry-phase (geometric phase) supported topological transport theory in inversion and time-reversal invariant nodal-line semimetals. In addition to transport properties, the team also focuses on topological band degeneracies in nodal-line semimetals and provides a classification map based on this [33]. A complete taxonomy of nonsymmorphic (non-comparable by symmetry) band degeneracies in hexagonal materials with strong spin-orbit coupling is elaborated upon. (Nonsymmorphic symmetries are symmetries that not comparable by the usual symmetry properties because symmetry operations do not have a common point on the lattice.)

Floquet engineering (reshaping the band structure) is another interesting experimental method, treating nodal topological semimetals as linked chains. There is malleability in periodic systems such that if the nodal rings are driven to Floquet Weyl points, the drumhead surface states become Fermi arcs [24]. The nodal rings allow the formation of nodal chains, in a linking structure (nodal-link semimetals). A nodal-link semimetal is a flexible structure to which an external periodic field can be applied to produce a Floquet Hopf insulator. As well as being of practical utility, nodal-link semimetals could support the development of topological field theories in the Brillouin zone.

### 2.2.3. Magnetic Topological Semimetals

Magnetic transport properties and magnetic phases comprise a growing research area in topological semimetals. The interacting nature of magnetic materials has been more challenging to address than dynamics in noninteracting systems. One line of research investigates topological quantum paramagnets, exotic states of matter whose magnetic excitations indicate a topological band structure. (A paramagnet is a material that is weakly attracted by external magnetic fields.) Ongoing work demonstrates that topological spin excitations can exist in the quantum-disordered paramagnetic phase of a spin ladder [34] and on a honeycomb lattice [35]. The honeycomb lattice bilayer structure is able to host a time-reversal symmetry-protected topological quantum paramagnetic phase. Of note is that the system can be induced to undergo a quantum phase transition in which the triplet-state edge modes become detached from the bulk excitations and are protected

by both a chiral and a unitary symmetry. The result might have practical applications in Mott insulators (devices that insulate rather than conduct at low temperatures per electron–electron interactions not considered in traditional band theory).

A classificatory schema for magnetic topological materials has also been developed, attempting to incorporate all magnetic symmetry group representations and topology [36]. The taxonomy treats various semimetal compounds and their synthesis methods, in various topology types (Weyl, Dirac, and nodal-line semimetals), for different classes of magnetism (antiferromagnetic, ferromagnetic, and non-collinear).

### 2.2.4. Chiral Topological Semimetals

Chiral topological semimetals are semimetals in which chirality or chiral anomaly plays a role [37]. Such chiral topological semimetals (Weyl fermions and multifold fermions) are interesting because they have crystal symmetries which can protect band crossings with degeneracies at high-symmetry points (multifold crossings). One project investigated optical conductivity in the crystal structure of a chiral topological semimetal (CoSi) which hosts multifold fermions (quasiparticles) [38]. The chiral crystals exhibit nonsymmorphic symmetries (a lack of inversion and mirror symmetry) that can be used to realize multifold crossings. These kinds of multifold fermions have been shown to exist in various chiral semimetals, which adds to the slate of manipulable topological semimetal materials.

Another project investigated topological semimetals in crystals in more detail [39]. Chiral crystals have handedness due to a lack of mirror and inversion symmetries and are expected to exhibit exotic phenomena such as long Fermi arc surface states, fermionic excitations, and unusual magneto-transport and lattice dynamics properties. However, the initially confirmed topological semimetals are in crystals that have mirror operations, thus prohibiting the exotic properties. A different material, however, AlPt, is demonstrated as an example of a chiral topological semimetal that does host multifold fermions (new forms of fourfold and sixfold fermions), which can be seen as a higher-spin generalization of Weyl fermions that does not appear in elementary particle physics. The multifold fermions are located at high-symmetry points and have Chern numbers (topological invariant metric) larger than those in Weyl semimetals, producing multiple Fermi arcs that span the diagonal of the Brillouin zone. The long Fermi arcs are experimentally imaged to determine the magnitude and sign of the Chern number.

## 3. Long-Range Entangled Topological Materials

### 3.1. Quantum Hall States

Quantum Hall states are a form of topological order that falls outside Landau symmetry breaking. In particular, the "fractional quantum Hall effect" is of interest to fault-tolerant quantum computing, and is obtained by applying a strong magnetic field perpendicular to a two-dimensional electron system at low temperature. In fractional quantum Hall states, electrons create quasiparticles (collective states) that have a fraction of the charge of a single electron and obey anyonic statistics [40].

The Hall effect is the production of a voltage difference (Hall voltage) across an electrical conductor that is transverse (perpendicular) to an electric current in the conductor and to an applied magnetic field perpendicular to the current (per Edwin Hall 1879). The idea is that a current of electrons in a thin conducting strip (two-dimensional plane) is subject to a constant magnetic field in the normal direction, while the Lorentz force perpendicular to the current causes a buildup of charge on the edge of the strip that induces a voltage across the width of the strip. The quantum Hall effect is the quantum version of the Hall effect, observed in two-dimensional electron systems at low temperature as magnetic fields are applied and the Hall conductance takes on quantized values.

The "fractional quantum Hall effect" indicates quantized plateaus at fractional values of charge, giving rise to quasiparticles (collective states) in which electrons bind magnetic flux lines to make new quasiparticles that have a fractional charge and obey anyonic statistics [5,6]. (The 1998 Nobel prize was awarded for the discovery of a new form of

quantum fluid with fractionally charged excitations.) The integer quantum Hall effect indicates quantized plateaus at integer values of charge. The result is quantized tiers ("Hall plateaus") that persist when electron density is varied; there is a finite density of states that are localized (pinned, as in the Anderson localization), which is useful in computational devices as electrons can be pinned (localized). A further advance is quantum spin Hall states, instantiating the quantum Hall effect based on the flow of spin currents (as opposed to charge currents), for potential application to next-generation quantum computing methods. A matrix mechanics formalism (allowing the ability to diagonalize multiple matrices to aid in solving many-body problems) has been extended to quantum Hall states, to characterize entanglement and emergent structure [41].

*3.2. Quantum Spin Liquids*

The main way that long-range entanglement is engaged in quantum matter systems is with quantum spin liquids. Quantum spin liquids are quantum matter phases in which the elementary degrees of freedom are magnetic spins (which can be instantiated as qubits in a quantum computational system) [42]. In general, magnetic systems are ordered in one of three ways: with all spins pointing in the same direction (ferromagnet, as a refrigerator magnet), disordered with neighboring spins (on different sublattices) pointing in opposite directions (antiferromagnet), or a frustrated order that is a combination of both (spin liquid or spin glass). Quantum spin liquids are the quantum version of a spin liquid, a "liquid" of disordered spins, that, is a phase of matter formed by quantum spins interacting in magnetic materials. Specifically, a quantum spin liquid is a magnetic system that does not settle into a large-scale ordered configuration, even at zero temperature, and resides in a nontrivial quasi-disordered ground state, which can be manipulated. Quantum spin liquids are typically characterized by topological order, long-range entanglement, and fractionalized (anyon) excitations.

In the usual situation of regular magnets at low temperature, electrons stabilize and form large-scale patterns (such as domains, stripes, or checkerboards) with magnetic properties. However, in a quantum spin liquid, the electrons do not stabilize when cooled and preserve their disorder much in the way liquid water exists in a disordered state. The electrons are constantly changing and fluctuating (like a liquid) in a highly entangled quantum state. Hence, the quantum spin liquid is called a liquid because, like a liquid, the fluctuating elements (electrons) do not settle into in a regular lattice as in a solid. Quantum spin liquids are attractive in quantum computing for the possibility of creating topological qubits made with quantum spin liquid matter phases (by instantiating the quantum spin liquids in a geometrical array).

3.2.1. Initial Discovery of Quantum Spin Liquids

Several different physical models have a disordered ground state that can be described as a quantum spin liquid. The real-life mineral, Herbertsmithite (named after mineralogist Herbert Smith), was discovered in Chile in 1972, and subsequently in many other locations (Iran, Chile, Arizona, and Greece). Herbertsmithite is a mineral with quantum spin liquid magnetic properties (neither ferromagnet nor antiferromagnet). The magnetic particles of the material have constantly fluctuating, scattered orientations in a kagome (triangle–hexagon) lattice. The mineral is comprised of Zinc, Copper, Oxygen, Hydrogen, and Carbon. In the laboratory setting, a specific kind of proposed quantum spin liquid formulation, the Kitaev honeycomb, was measured experimentally in 2015 with the excitation of a spin liquid on a honeycomb lattice with neutrons in a graphene-like material (ruthenium) (Oak Ridge National Laboratory [43]). Measurements confirmed the expected properties of the quantum spin liquid, namely, strong spin orbit coupling and low-temperature magnetic order.

### 3.2.2. Creating Quantum Spin Liquids from Scratch

In 2021, two projects created quantum spin liquids from scratch, specifically demonstrating the long-range entanglement property, by using a coupled superconducting circuit and an optical atom array [44]. The former team (from Google Quantum AI) used a 32-qubit quantum processor to study the ground state and excitations of the toric code [45]. The latter team (from the Lukin laboratory at Harvard) detected signatures of a toric code-type quantum spin liquid in a two-dimensional array of Rydberg atoms held in optical tweezers (lasers) [46]. The central achievement for both projects was engineering the topological order known as the toric code, an archetypical two-dimensional lattice model that exhibits the exotic properties of topologically ordered states and is proposed for quantum error correction [2].

In more detail, the first team realized topologically ordered states using a 32-qubit superconducting quantum processor (Sycamore). The ground state of the toric code Hamiltonian was prepared using an efficient quantum circuit on the Sycamore processor. The topological nature of the state was experimentally established by measuring the topological entanglement entropy (topology-based measure of quantum entanglement entropy) and by simulating anyon interferometry to extract the braiding statistics of the emergent excitations. The second team used a 219-atom programmable quantum simulator to probe quantum spin liquid states. Arrays of atoms were placed on the links of a kagome lattice (lattice comprised of equilateral triangles and hexagons). The onset of the toric-type quantum spin liquid phase was detected using topological string operators (which indicate the signatures of topological order and quantum correlations). A class of dimer models (molecules with identical molecules linked together) was implemented as a promising candidate to host quantum spin liquid states.

One result of the quantum spin liquid demonstrations is new understandings of the bulk–boundary relationship in condensed matter physics. The bulk–boundary relationship usually entails some range of unrestricted boundary behavior, within the context of boundary symmetries that are linked to and protected by bulk invariants. However, having an experimental platform for the more rigorous creation and manipulation of quantum spin liquids is revealing new things such as that, strikingly, under certain open boundary conditions, the boundary itself undergoes a second-order quantum phase transition, independent of the bulk [47]. Future work could tackle creating even more precise atomic quantum spin liquids, assembled from scratch by building lattices of magnetic atoms from the bottom up, literally atom-by-atom, with the probe tip of a scanning tunneling microscope positioning the atoms on the surface [48].

### 3.2.3. Topological Qubits, Non-Locality, and Quantum Error Correction

The reason quantum matter phases with long-range entanglement are attractive as potential topological qubits in quantum computing is due to the error-correction possibility afforded by non-locality. Working with quantum spin liquid phases entails accessing non-local observables, through, for example, topological string operators. The non-local nature of quantum spin liquid states makes them attractive platforms for fault-tolerant quantum computation, as quantum information encoded in locally indistinguishable ground states is robust to local perturbation. The principle underlying topological quantum error-correcting codes is that, in the quantum spin liquid model, the logical codespace corresponds to the degenerate ground state subspace of a lattice model. The key benefit of long-range entanglement is being able to perform quantum error correction (through non-local measurements).

### 3.3. Entanglement Entropy and Quantum Phase Transition

Entanglement is an important aspect of being able to manage system criticality and phase transition. A phase transition between different quantum phases can be triggered by a change in physical parameters such as magnetic field or pressure [49]. Whereas classical phase transition is triggered by varying a macroscopic physical parameter such as temperature or density, quantum phase transition is caused by changing a quantum physical

parameter at zero temperature, tuning a non-temperature variable such as magnetic field, pressure, or chemical composition (via a Hamiltonian term). Although a classical phase transition is often temperature-based (often called thermal phase transition), a quantum phase transition is not carried out by varying the temperature, because the system remains at zero temperature throughout; the quantum system is at zero temperature before and after the phase transition.

### 3.3.1. Topological Entanglement Entropy

Topological entanglement entropy is a topology-based measure of entanglement entropy specific to quantum matter and quantum phase transition. The phases on either side of a quantum critical point may be characterized by different kinds of topological order. The quasiparticle excitations or quantum correlations among the microscopic degrees of freedom might have qualitatively different properties in the two phases. Since it may not be possible to distinguish the phases based on a local order parameter, a non-local parameter is needed such as a topology-based measure of the long-range system entanglement.

Topological entanglement entropy incorporates aspects that the usual quantum entropy measures (e.g., von Neumann entropy and Rényi entropy) do not, that are specific to measuring entanglement entropy in quantum many-body states with topological order [50]. The topological entanglement entropy is calculated either from the quasiparticle excitations of the many-body state or in a comparison between the system and the von Neumann entropy. (Specifically, topological entanglement entropy is computed as the logarithm of the total quantum dimension of the quasiparticle excitations of the state, or by comparing the von Neumann entropy between a spatial block and the rest of the system).

### 3.3.2. Topological Quantum Field Theory

In fact, since it treats topological invariance, topological entanglement entropy constitutes a topological quantum field theory, in three dimensions (two space and one time), in the general formulation. A topological quantum field theory is a quantum field theory that emphasizes topological invariants and in which the correlation functions do not depend on the metric of spacetime. This means that the theory is not sensitive to changes in the shape of spacetime; if spacetime warps or contracts, the correlation functions do not change, and they are topologically invariant. Just as any topological object which can bend and be deformed but not cut and the invariant properties persist, so too in a topological quantum field theory, spacetime can warp or contract but the correlation functions do not change and remain topologically invariant. This topological entanglement entropy formulation is operationalized as tripartite information (equations involving two space and one time dimension) [51].

## 4. Codes (Toric, etc.)

A code is a mathematical model for encoding a quantum system. In the quantum matter context, codes have various purposes. First is for the general characterization and solvable modeling of quantum systems. Second is to confirm well-formedness conditions in the realization of synthetic quantum matter (as demonstrated by the quantum spin liquid projects [45,46]). Third is to error-correct the position of atoms and other quantum objects in the realization of quantum computation.

### 4.1. Stabilizer Codes

#### 4.1.1. The Toric Code

A central formulation developed by Kitaev is the toric code, which is a solvable model for studying highly entangled quantum phases (of which topological matter phases are the primary example) [2]. The toric code model considers spin-1/2 spins in a square lattice, with operators (plaquette and star) calculated as products over the spins on the bonds [52]. The operators commute, which allows the ground state and other energy eigenvalues to be computed. The Kitaev honeycomb lattice model extends the toric code model to lattices

and provides a simple Hamiltonian with nearest-neighbor interactions between spin-1/2 spins with an exact solution (and a stable gapless quantum spin liquid phase) [53]. The toric code is an example of the class of stabilizer codes based on commuting operators. Other codes such as subsystem codes are based on non-commuting operators (Bacon–Shor code and free-fermion codes [54]).

### 4.1.2. Quantum Error Correction

An important use of codes is for error correction in quantum computing. Quantum systems are much more sensitive to environmental noise than classical systems, and particularly as they evolve through operations, errors arise which need to be corrected. Quantum systems are therefore designed with both the physical elements (atoms, molecules, spins) that conduct the processes, and a much larger set of elements (ancilla) to correct the errors. Known classical error-correction methods such as making redundant copies or checking information integrity before transmission are not possible in quantum systems since information cannot be copied or inspected (per the no-cloning and no-measurement principles of quantum mechanics). Quantum error correction therefore often relies on entanglement instead of redundancy. The quantum state to be protected is entangled with a larger group of states (the ancilla) from which it can be corrected indirectly. Various quantum error-correction codes are listed in Table 2.

**Table 2.** Codes for Quantum Matter Characterization, Creation, and Manipulation.

| | Code | Description |
|---|---|---|
| 1 | Code (general) | Allowed values (or value structure) for data or other parameters |
| 2 | QEC code | Logical codespace corresponding to a physical lattice model space |
| 3 | Stabilizer code | Topology-based Pauli operators (X, Y, Z) to restore bit/spin flip |
| 4 | Toric code | Stabilizer codes defined on a 2D torus-shaped spin lattice |
| 5 | Surface code | Stabilizer codes defined on a 2D spin lattice in any shape |
| 6 | Bosonic code | Self-contained photon-based oscillator system with bosonic modes |
| 7 | GKP code | Bosonic code: squeezed states protect position-amplitude shifts |
| 8 | Molecular code | GKP codes extended to asymmetric bodies (molecules) in free space |
| 9 | Cat code | Superpositioned states (Schrödinger) used as error-correction codes |

The standard errors are a bit flip, a sign flip (the sign of the phase), or both. Basic codes diagnose the error, which correspond to Pauli matrices for controlling the qubit (or spin) in the X, Y, and Z dimensions. The error is expressed as a superposition of basis operations given by the Pauli matrices. If there is an error, the same Pauli operator is applied to act again on the corrupt qubit to reverse the error effect. The unitary correction returns the state to the initial state without measuring the qubit directly. The physical state of the qubit (spin, atom, photon, molecule) is literally controlled by using its physical properties such as rotational state to realign it [55].

### 4.1.3. Stabilizer Codes in Quantum Error Correction

The basic quantum error-correcting code is the stabilizer code, as the quantum version of linear codes used classically. The stabilizer code (mainly applied through the toric code and the surface code) is a topology-based method that interprets particle movement and its correction in the structure of lattice topologies. The toric code is instantiated as a stabilizer code defined on a two-dimensional lattice with periodic boundary conditions (thus giving the shape of a torus), with stabilizer operators on the spins around each vertex and plaquette (face). Surface codes are a more generic formulation of stabilizer codes, also defined on two-dimensional spin lattices, and take various shapes but are not necessarily toroidal. Lattice surgery is a method of switching between codes on the fly.

*4.2. Bosonic Codes*

Stabilizer codes as the traditional approach to quantum error correction is a mechanism that requires multiple qubits in two-tier registers, which is cumbersome and, as systems scale, can give rise to even more errors. A more compact method is encoding the qubits in a self-contained (continuous variable) system such as bosonic modes (photon states), in the form of a harmonic oscillator [56]. A generalized form of bosonic codes is GKP codes (Gottesman, Kitaev, Preskill) which conduct error-corrected qubit encoding in an oscillator using superpositions of squeezed states (quantum noise-reduced oscillatory states) to protect against shifts in position and amplitude, applied based on the grid states of an oscillator [57]. Superpositioned qubits for error correction are generally known as (Schrödinger) cat codes or cat states [58].

GKP codes correct errors (seen as molecular displacement) by controlling the position and momentum of an oscillator with known symmetric rotations. Molecular codes extend GKP codes by allowing rotations to be performed on asymmetric rigid bodies in free space, in quantum systems ranging from oscillators to diatomic and polyatomic molecules [59]. Error-corrected molecular control is an important capability in quantum circuit design and quantum nanoscience.

## 5. Application Example: Quantum Neuroscience

This section provides an example of how the quantum matter concepts and methods discussed in this review may be applied to other fields, in particular, quantum neuroscience as a complex and emerging quantum studies field. Quantum matter, studying the foundations of matter, is uncovering some of the most fundamental physical formulations possible, and these articulations might be widely applicable to the modeling of complex phenomena in quantum computational environments such as neuroscience.

*5.1. Quantum Neuroscience*

Quantum neuroscience is an emerging field in quantum biology which studies potential quantum effects in the brain and applies quantum information science methods to problems in neurobiology. The field is proceeding along three lines of activity in the analysis of wavefunctions, neural dynamics, and neuroscience physics (neuroscience interpretations of foundational physics findings). In quantum neuroscience, neural behavior is modeled based on the quantum-mechanical properties of superposition, entanglement, and interference, which could be extended with quantum matter findings. Neuroscience physics approaches in quantum neuroscience already incorporate the hyperbolic space of the AdS/CFT correspondence and the notion of successive bulk–boundary tiers. These methods might be further enhanced with the formulations of quantum matter systems which analyze short-range symmetry protection and long-range entangled systems in which non-local measurements are possible.

5.1.1. Neural Signaling Phase Transitions

The complexity of the human brain (operating at nine orders of magnitude scale tiers ranging from the central nervous system to ion channels [60]) suggests the application of physical models that allow system manipulation across scales through the foundational properties of topology, symmetry, and geometry. Neural signaling is often cast as a phase transition problem, for which topological entanglement entropy might be applied to model the different sides of the phase transition. Quantum models are needed to incorporate synaptic integration (aggregating thousands of incoming spikes from dendrites and other neurons) and the electrical–chemical signaling of neuron–glia interactions at the molecular scale. These kinds of neural signaling integration problems require a higher level of sophistication than has been available classically, including, for example, the ability to solve partial differential equations [61].

5.1.2. AdS/Brain Theory (Anti-de Sitter Space)

The AdS/Brain theory has been proposed as a multiscalar theory of neural signaling based on the AdS/CFT correspondence that incorporates the four scale tiers of network, neuron, synapse, and molecule [62,63]. The AdS/CFT correspondence (Anti-de Sitter space/Conformal Field Theory) is a theory positing that a physical system with a bulk volume can be described by a boundary theory in one fewer dimensions [64]. Multi-tier bulk–boundary pairings are implicated in the neural signaling operation, each a bulk to the other's boundary (network–neuron, neuron, synapse, and synapse–ion). Entanglement is measured and renormalized across the system with bMERA (brain) random tensor networks evolved with Floquet periodicity-based neural dynamics. A matrix quantum mechanics formulation is suggested to diagonalize multiple matrices, and the neural signal can more expediently search through the signaling space with quantum walks.

The quantum matter program enables a number of enhancements to the AdS/Brain theory as follows (Table 3). The bulk–boundary relationship elucidations in topological semimetals formalizing how bulk wavefunctions lead to various manipulable symmetry-protected surface states can be incorporated [25], in both an electronic spin and magnetic models [35]. An integration of the hyperbolic Bloch theorem, hyperbolic band theory, and hyperbolic lattices is indicated [11], as well as the long-range entanglement handling in quantum spin liquid systems [45,46]. Quantum matter–quantum information science models offer new possibilities for realizing the quantum circuits for modeling neural signaling with MERA-based wavefunctions using classical platforms (TPUs) [65]. (MERA, the multi-scale entanglement renormalization ansatz, is a tensor network representation for ground states of critical quantum spin chains.) The Floquet periodicity in neural signaling could be modeled with the advance in Floquet engineering techniques available on existing quantum computational platforms [66]. A new quantum matter-based matrix mechanics model that incorporates the quantum Hall effect could allow the multi-dimensional scalability required by the AdS/Brain theory [41]. The real-life behavior of neural signaling could be further instantiated per molecular code rotational control in quantum matter systems [59]. Finally, a research program applying Chern–Simons theory to the biological context (topological curvature indicates max–min points) [67] could be integrated with the chiral topological semimetal formulations in which high Chern numbers quantify topological invariance [39].

**Table 3.** Apply Quantum Matter Concepts and Methods to Extend the AdS/Brain Theory.

|   | Model Parameter | Quantum Information | Quantum Matter |
|---|---|---|---|
| 1 | Multiscalar model | AdS/CFT correspondence | Condensed matter bulk–boundary |
| 2 | Phase transition | Neural signal integration | Topological entanglement entropy |
| 3 | Symmetry | Symmetry rebalancing | Short-range: topological semimetal |
| 4 | Entanglement | Non-local measurement | Long-range: quantum spin liquid |
| 5 | Renormalization | bMERA tensor network | MERA (wavefunction) TPUs |
| 6 | Floquet dynamics | Neural signal periodicity | Floquet circuits with NISQ QC |
| 7 | Matrix mechanics | Diagonalize matrices | Quantum Hall matrix mechanics |
| 8 | Quantum walks | Faster lattice search | Molecular code rotation |
| 9 | Optimization | Chern–Simons curvature | High Chern numbers (invariance) |

*5.2. Quantum Biology of Deformable Soft Solids*

Another way that quantum matter approaches may be relevant in the study of biology is related to the topological aspects of soft deformable materials. Amorphous solids such as tissues, foams, and emulsions are composed of deformable particles that act together in cooperative behaviors such as cell motility. The impact of single-particle deformability on the collective behavior of soft solids is not well-understood. Research finds that the packing mechanism is different for soft deformable particles than for rigid-shape particles, involving more degrees of freedom (parameters), and appears to be influenced by collective vibrational and mechanical properties [68].

## 6. Discussion

This paper provides an overview of activity in the quantum matter field (quantum materials with topological order or topological matter phases). Research is described in the areas of short-range protected topological materials (particularly topological semimetals), long-range entanglement in quantum Hall states and quantum spin liquids (which allow quantum error correction through non-local measurements), and codes (quantum system encodings for the characterization and control of quantum matter systems). An applied example in quantum neuroscience is considered. The fast pace of research is leading to a larger conceptualization of the field of quantum matter in two important ways. First is an appreciation that not only are topology and symmetry important for characterizing materials, but also geometry [69]. Second, the initial focus on short-range topologically protected materials is expanding to also include the analysis of long-range entangled materials [70].

There are many potential risks and limitations to this work. The review is intended to highlight representative projects in quantum matter science and necessarily omits others. A subsequent level of detail is available in other reviews of quantum matter [71–73] and quantum spin liquids [52,74]. The topological matter industry experienced a minor setback in the 2021 retraction of a high-profile *Nature* paper [75]. The paper (from researchers at Delft University of Technology in the Netherlands, later associated with Microsoft) claimed to have found the definitive signature of Majorana zero-mode quasiparticles, but results could not be replicated. The incident appears isolated and has not diminished the brisk pace and variety of approaches in topological matter.

### 6.1. Quantum Matter and Quantum Information Science Integration

#### 6.1.1. Universal Quantum Computing

The potential importance of quantum matter is that the field is establishing foundational developments that could lead to universal fault-tolerant quantum computing. Existing quantum computing methods are early demonstrations of the platform, but substantial technical breakthroughs in quantum error correction are needed to progress from NISQ (noisy intermediate-scale quantum) devices to fully FTQC (fault-tolerant quantum computing) [76]. Topological formulations offer potential solutions for a new class of fault-tolerant universal quantum computation, thus integrating the fields of quantum matter and quantum information science (Figure 1).

#### 6.1.2. Quantum Machine Learning

Digital methods such as quantum machine learning are proving indispensable to many fields. Quantum matter provides a rich environment for the implementation of machine learning techniques, including innovations specific to the quantum matter context. Artificial neural networks are used to simulate strongly correlated systems, overcoming limitations inherent in the usual classical and quantum Monte Carlo methods [77]. The methods are extended in a probabilistic model to simulate quantum many-body dynamics (using factorized generalized measurements to map quantum states to probability distributions to obtain an exact formulation of quantum dynamics) using a transformer neural network (state-of-the-art natural language-processing model) [78]. Other teams apply variational methods to quantum many-body systems [79], and use quantum matter itself as a computational technology in symmetry-protected topological phases in a quantum cellular automata model [80].

#### 6.1.3. Quantum Simulation Platforms

Tabletop platforms for the realization of quantum phenomena have been an important step in advancing their scientific study. In quantum matter, circuit quantum electrodynamics (cQED) provides a means of analyzing and synthesizing quantum matter phases in hyperbolic space [9]. Circuit quantum electrodynamics joins other experimental tabletop platforms such as chip-based particle accelerators [81], black hole on a supercon-

ducting chip [82], and quantum gravity in the lab setups (via Rydberg atom arrays and trapped ions) [83].

### 6.1.4. Quantum Nanoscience

Quantum matter may be synthesized with quantum simulation platforms, and more formally in the area of quantum nanoscience, which is concerned with fabricated nanostructures that exploit quantum properties. Quantum nanoscience actively incorporates quantum properties in device design rather than managing around them, as progress in nanoscience has enabled precision control in the fabrication and synthesis of matter [84]. A quantum nanoscience roadmap is set forth in the areas of quantum computing, sensing, and communications [85]. Key areas include quantum dots (semiconductor nanoparticles with quantum-mechanical optical and electrical properties), plasmonics (concentrated optical energy), single-molecule and single-electron transistors, levitated nanoparticles (controlled with optical tweezers), as well as the topological insulators and quantum spin liquids discussed in this review.

### 6.2. Future Outlook for Quantum Matter Science

Quantum matter is a foundational science supporting the development of other quantum sciences fields (ranging from quantum chemistry to quantum astronomy [86]), as well as in the application of quantum methods to humanities studies, as the quantum humanities may extend the digital humanities [87,88]. More speculatively in the further future, quantum matter-inspired Floquet engineered pocket devices might not be out of the question, for example, if near-field optics were to replace lasers as a more efficient field generator, in a new era of consumer electronics based on "metamaterial plasmonics" [89].

**Author Contributions:** Conceptualization and writing; M.S.; review and suggested additions; R.P.d.S., F.W. All authors have read and agreed to the published version of the manuscript.

**Funding:** This research received no external funding.

**Conflicts of Interest:** The authors declare no conflict of interest.

### Abbreviations

| | |
|---|---|
| Abelian | Commuting (order of terms does not matter) |
| AdS/Brain | Multi-tier holographic theory of neural signaling based on the AdS/CFT correspondence |
| AdS/CFT correspondence | (Anti-de Sitter space/Conformal Field Theory) Theory positing that a physical system with a bulk volumecan be described by a boundary theory in one fewer dimensions |
| Angular momentum | Measure of a body's tendency to roll, spin, or orbit; measured by amount (magnitude), direction (projection), and intrinsic angular momentum (spin) |
| Anyon | Third type of particle between a fermion and a boson, not fundamental but emerges in many-body systems |
| Anyonic exchange statistics | Computable measure of particles changing places, exchanging wavefunctions by 'braiding' |
| Artificial lattice | Atomic-scale structure designed to confine electrons or spins on a chosen lattice (made with scanning tunneling microscopy or electron beam) |
| Band gap | Prohibited range of energy in a system |
| Berry curvature | Gauge-invariant geometrical property of a band; invariant under changes in the phase of the wavefunction |
| Bloch theorem | Solution to the Schrödinger equation for periodic systems |
| Boson | Force particle (photon, gluon, graviton) with integer spin |

| | |
|---|---|
| Bosonic code | Photon-based system in which physical and protective logical qubits can be encoded in a self-contained continuous value system in a single bosonic mode (state) |
| Bosonic mode | Photon state controllable with standard Gaussian operations such as squeezing, displacement, phase rotation, and beam splitting |
| Brillouin zone | Geometric zone inside a reciprocal lattice or crystal |
| Cat code | (Schrödinger's cat) error correction based on superpositioned coherent states |
| Charge | Electrical valence of a particle (particle properties: spin, charge, angular momentum) |
| Chern number | Topological invariant of the Berry curvature flux over a closed momentum surface (global system property) |
| Chern–Simons theory | Mathematical model of topological invariance |
| Chiral topological semimetals | Topological semimetals in crystals with a chiral structure (handedness due to lack of mirror and inversion symmetries) |
| cQED Circuit quantum electrodynamics | Experimental platform for modeling and synthesizing quantum matter phases in hyperbolic space |
| Code (error correction) | Allowed values (or structure of values) for data in a system, may include parameters re: how many ancilla (extra) bits protect one logical bit over what distance |
| Codespace | Error-correction domain; possibly denoted by lattice grid states or graph states |
| Coherent state | Oscillatory quantum state (the quantum state of the harmonic oscillator) |
| Correlation function | The average (expectation value) of field operators at different positions; the amplitude for propagation of a particle or excitation between two points |
| Crystal | Atoms organized in a regular array (lattice); has discrete translation symmetry |
| Crystal lattice | Symmetrical three-dimensional arrangement of atoms inside a crystal |
| Dimer | Molecule of two identical molecules linked together |
| Discrete time crystal | Non-equilibrium state of matter that breaks time translation symmetry (repeating time structure) |
| Eigenvalues | Values at allowable scale tiers in a system, levels; characteristic system values |
| Electron holes | Positively charged quasiparticles denoting the lack of an electron in a state in the valence band of a semiconductor |
| Energy band theory | Allowed/prohibited energy bands/band gaps in systems |
| Entanglement | Quantum property of correlated physical attributes among particles (position, momentum, spin, polarization) |
| Fermion | Matter particle (electron, quark) with half-odd integer (1/2, 3/2, etc.) spin |
| Fractional quantum Hall effect | Quantized plateaus at fractional values of charge, giving rise to quasiparticles (collective states) in which electrons bind magnetic flux lines to make new quasiparticles that have a fractional charge and obey anyonic statistics |
| Floquet engineering | Control of periodically driven time cycles in quantum matter systems |
| Floquet theorem | Periodic system (Bloch) transform to solvable linear differential equations |
| Frontier orbitals | Highest occupied/lowest unoccupied orbitals of a molecule |
| Fuchsian model | Hyperbolic Riemannian surface model |

| | |
|---|---|
| Gapless | No band gap in spacing between energy levels in a system |
| Gapped | Gaps in energy bands in a system |
| GKP codes (Gottesman, Kitaev, Preskill) | Error-corrected qubit encoding in an oscillator with superpositions of squeezed states, protected against shifts in position and amplitude damping |
| Hall effect (Hall conductance) | Production of a voltage difference (the Hall voltage) across an electrical conductor that is transverse (perpendicular) to an electric current in the conductor and to an applied magnetic field perpendicular to the current (Hall 1879) |
| Hamiltonian | Operator (function) used to calculate the energy levels of a quantum system |
| Herbertsmithite | Mineral with quantum spin liquid magnetic properties (neither ferromagnet nor antiferromagnet); magnetic particles with constantly fluctuating, scattered orientations on a kagome lattice; (Zinc, Copper, Oxygen, Hydrogen, Chlorine); (Iran, Chile, Arizona, Greece) |
| Hilbert space | Infinite-dimensional space of quantum mechanics (vs 3D Euclidean space) |
| Honeycomb lattice | Standard hexagonal/triangular lattice (e.g., graphene) (generally looks the same from any direction) |
| Hyperbolic band theory | Energy band theory in hyperbolic space |
| Hyperbolic lattice | Synthetic quantum matter in which particles hop on a discrete tessellation of two-dimensional hyperbolic space |
| Hyperbolic space | Geometric space with negative curvature (vs Euclidean space (zero curvature) and elliptic space (positive curvature) |
| Josephson junction | Quantum tunneling superconducting device used in quantum computing |
| Josephson junction-based superconducting circuits | Superconducting qubits controlled with microwave photons (quantized electromagnetic fields stored in the superconducting circuits) |
| K-space | Wave vector space of possible values of momentum for a particle (also the spatial frequency domain of a Fourier transform or a compactly generated topological space) |
| Kagome lattice | Uniform tiling of equilateral triangles and hexagons |
| Kitaev honeycomb lattice | Exactly solvable spin model in two dimensions; spins are on the vertices of a honeycomb lattice with nearest-neighbor interactions |
| Lattice | Regular array geometric arrangement of matter in a space (e.g., crystal); scaffolding |
| Lattice surgery | Switching between error-correcting codes on the fly |
| Magnon | Collective excitation of electron spins in a crystal lattice |
| Many-body problem | Physical systems with many interacting particles (three/more) |
| Majorana fermions | Exotic fermions that are their own antiparticles |
| Many-body localization (MBL) | Many-body interactions causing quantum particles to be localized and maintained in an out-of-equilibrium state |

| | |
|---|---|
| Multi-scale entanglement renormalization ansatz (MERA) | Tensor network representation for ground states of critical quantum spin chains (with a network that extends in an additional dimension corresponding to scale) |
| Mode | Characteristic state, normal frequency, allowable value |
| Molecular code | GKP codes for asymmetric bodies (molecules) in free space |
| Nodal-line semimetals | Topological semimetals with energy band-touching manifolds in the shape of closed loops |
| Non-abelian | Non-commuting (order of terms matters) (v. abelian) |
| Nonsymmorphic | Not comparable on a symmetry basis (symmetry operations do not have a common point on the lattice) |
| Paramagnet | Material weakly attracted by an external magnetic field |
| Particle properties | Spin, charge, angular momentum, polarization |
| Phonon | Collective excitation of atoms in a rigid crystal structure |
| Plasmon | Collective excitation of electrons simultaneously oscillating with respect to ions |
| Quantum droplet | Quantum matter phase defined by properties emerging from the interactions of bosonic or fermionic constituents |
| Quantum error-correcting code | Logical codespace corresponding to the physical subspace of a lattice |
| Quantum Hall effect | Quantum version of the Hall effect; obtained by applying a strong magnetic field perpendicular to a two-dimensional electron system |
| Quantum matter | Novel phases of matter at zero temperature with emergent order and exotic properties, possibly including the emergence of quasiparticles (collective excitations) with anyonic exchange statistics, gauge theory, quantum phase transitions, and low-energy effective theories of topologically ordered states |
| Quantum nanoscience | Nanostructure fabrication that exploits quantum effects |
| Quantum phase transition | Phase transition between different quantum phases via parameter change such as magnetic field or pressure |
| Quantum spin Hall states | Quantum Hall effect based on the flow of spin currents (as opposed to charge currents) |
| Quantum spin liquid | Quantum matter phase with magnetic spins (qubits) degrees of freedom; a magnetic system that does not settle into a long-range ordered configuration, even at zero temperature, residing in a nontrivial quasi-disordered ground state |
| Quantum topology | Novel properties of topological shapes in quantum systems |
| Quasiparticle | Long-lived, low-energy excitation of a many-body state in fermions (collective excitations in bosons); examples: electron holes, phonons, plasmons, magnons |
| Rényi entropy | Composite of Shannon, Hartley, collision, and minimum entropy |
| "Schrödinger cat" states | Superpositioned quantum states |
| Soliton | Stable solitary wavepacket in nonlinear systems |
| Spin | Intrinsic form of angular momentum carried by elementary particles (depicted as an axis of rotation, but actual particles do not rotate); particles with spin may possess a magnetic dipole moment (exploited in electronic devices) |
| Spin chain | Linear collection of magnetic moments with spin–spin coupling interactions |

| | |
|---|---|
| Spin engineering | Control of spin systems in devices and materials, including the precise arrangement of magnetic atoms as a probe |
| Spinors | More complicated version of vectors and tensors needed to describe the rotations of particle spin |
| Spintronics (spin electronics) | Using electron spins as the internal degree of freedom to store 0 s/1 s for information processing |
| Squeezed (coherent) state | Pinched oscillatory quantum state (to reduce the quantum noise (environmental interference)) |
| Stabilizer code | Quantum error-correction code (quantum version of linear codes), based on X-, Y-, Z-axis Pauli operators to measure entangled states and correct (bit flip, spin flip) a corrupt quantum state to its original state; commuting operators |
| Subsystem code | Quantum error code with non-commuting operators |
| Surface code | Stabilizer code, topology-based, defined on a two-dimensional spin lattice, taking various shapes |
| Symmetry | Features of particles and spacetime which are unchanged under some transformation; property of looking the same from different points of view (face, cube, laws of physics) |
| Symmetry breaking | Phase transition, rupturing a system's symmetry (e.g., time-reversal, particle-hole, chiral) |
| Symmetry-protected topological (SPT) order | Quantum matter phases with trivial topological order (short-range entanglement), symmetry, and a finite energy gap (e.g., topological insulator) |
| Tessellation | Tiling of a plane using geometric shapes (tiles) |
| Time-reversal symmetry breaking | System property: the dynamics of a process remain well-defined when the sequence of time-states is reversed |
| Topological entanglement entropy | Topology-based measure of entanglement entropy specific to quantum matter and quantum phase transition calculated from the quasiparticle excitations of the many-body state or in a comparison between the system and the von Neumann entropy (tripartite information; two time, one space dimension) |
| Topological insulator | Material with a conducting surface and an insulating interior; surface states are symmetry-protected (e.g., time-reversal, particle-hole, chiral symmetry) |
| Topological quantum field theory | Quantum field theory emphasizing topological invariants and impervious to spacetime contraction; explains quantum matter phases |
| Topological qubits | Computational qubits made with quantum matter phases (e.g., by putting quantum spin liquids into a geometrical array) |
| Topological semimetals | Material with energy band-touching manifolds (at zero-dimensional points or one-dimensional nodal lines/rings) |
| Topological strings | Strings linking atoms entangled in a quantum spin liquid |
| Topology | The property of geometric form being preserved under deformation (such as bending, stretching, twisting, and crumpling, but not cutting or gluing) |
| Toric code | Stabilizer code defined on a two-dimensional lattice with periodic boundary conditions (torus-shaped); stabilizer operators on the spins around vertex and plaquette (face) |
| Trotterization | Operation to simulate the evolution of a Hamiltonian |

| Valleytronics (valley electronics) | Using valleys in the electronic band structure of the first Brillouin zone of a crystal as the internal degree of freedom to store 0 s/1 s for information processing |
| von Neumann entropy | (Quantum mechanical entropy) minimum over all measurement bases of Shannon entropy |
| Wavefunction | Quantum system state description (positions or speeds (momenta) of entire system configurations); generally intractable Schrödinger equation applied (complex-valued probability amplitudes with real and imaginary wave-shaped components) |
| Weyl and Dirac topological semimetals | Topological semimetals created by two energy bands crossing at a single node (Weyl node), with two-fold (Weyl) and four-fold (Dirac) degenerate Fermi points |

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
