# Peer review of "Quantum Matter Overview"

_2571-8800, doi:10.3390/j5020017_

Round 1

Reviewer 1 Report

The article represents an extensive review of the current state of research in the field of quantum matter, which will be of interest to a wide range of readers. The authors provide an extended description of the concepts and terms used to describe quantum properties. The article is written in clear scientific language and is well structured. I recommend the article for publication. 

Author Response

Thank you for the review

Reviewer 2 Report

This article is a collection of incoherent claims, most likely copied from other works with some text editing, glued together in a arbitrary way. The authors do not demonstrate any understanding of the subject and they are in fact not experts in the field. Reading the article does not help in any way to understand what quantum matter is and why it is interesting. Instead, reading this article is just wasted time and nothing more. The article contains a huge number of factual errors and false claims coming from a complete incomprehension of the subject. The article cannot be improved in principle. I do not recommend the publication of this article.

Author Response

Changes made per Reviewer 2 Comments

  1. The nature of posited claims was substantially reduced and off-topic material deleted, e.g.
    1. “The three-fold approach of characterization, digital simulation, and novel materials construction in quantum matter could become standard in research programs”
    2. “Quantum matter is a burgeoning field with far-ranging application not only in its own domain, but also as a concretized experimental platform for theoretical physics”
  2. Overall content topics and flow organized to be more coherent and in line with that in topological materials papers, starting with “short-range protected materials,” and then going into “long-range entangled materials”
  3. An attempt was made to be more explicit about “why quantum matter is interesting” by relating it to quantum information science and quantum computation with greater clarity

Reviewer 3 Report

A good woirk, I suggest some integrations, for ex:C.-K. Chiu, J. C. Y. Teo, A. P. Schnyder, S. Ryu, Rev. Mod. Phys. 88, 035005 (2016).
- D. G. Joshi, A. P. Schnyder, arXiv:1809.06387 (submitted).
- D. G. Joshi, A. P. Schnyder, Phys. Rev. B 96, 220405(R) (2017).
-Y.-H. Chan, C.-K. Chiu, M. Y. Chou, A. P. Schnyder, Phys. Rev. B 93, 205132 (2016).
- W. B. Rui, Y. X. Zhao, A. P. Schnyder, Phys. Rev. B 97, 161113(R) (2018).
- J. Zhang, et al., Phys. Rev. Materials 2, 074201 (2018).[1] C.-K. Chiu, J. C. Y. Teo, A. P. Schnyder, S. Ryu, Rev. Mod. Phys. 88, 035005 (2016).

Author Response

Thank you for the review. 

  1. All of these references have been added and discussed in a new section “2. Short-range Protected Topological Materials” that primarily reviews topological semimetals

Reviewer 4 Report

This review article gives a rough overview of the recent progress in quantum materials. The content dealt with is miscellaneous and not very up to date. To cover the very recent progress, I strongly suggest to include reviews on the following articles, which give extremely new concept of quantum material on hyperbolic space. 

https://arxiv.org/abs/2008.05489

https://iopscience.iop.org/article/10.1088/1361-648X/ac24c4

https://arxiv.org/abs/2107.10586

https://www.pnas.org/doi/10.1073/pnas.2116869119

https://arxiv.org/abs/2202.01538

Due to the development of quantum simulators, those new hyperbolic meta materials offer new possibilities for new quantum materials.

Author Response

Thank you for the review. 

  1. A “Hyperbolic Space” section and “Hyperbolic Bloch Theorem and Hyperbolic Band Theory” and “Magnetics” subcategories were added to discuss all of the suggested works
  2. Re: miscellaneous content. The paper has been recast more deliberately into “short-range protected topological materials” and “long-range entangled materials,” with several sections deleted (e.g. electron dynamics, spin engineering, Floquet engineering) to bring the work into greater conformity with conventional topical categories in topological materials study. Other sections were streamlined, for example “Quantum Droplets and Pre-droplet Solitons” was brought within the “Condensates” discussion

Reviewer 5 Report

The authors present a summary of activities in the quantum matter field. The manuscript is written in plain language and provides general information about quantum materials and their applications. Overall, this review is well written and organized. I’ll suggest the publication once the authors make minor changes listed below:

  1. Although this is a non-technical overview of activities in quantum matter for general audiences. But some figures which, for example, describe the relations between fields/physical phenomena/materials will be helpful for the audience to have a general idea about the field.
  2. Line 240: Wrong numbering.
  3. Line 275: Wrong numbering.

Author Response

Thank you for the review

  1. A chart has been added (Figure 1) to structure the material visually, along with a graphical abstract for the paper.
  2. The line-numbering has been revised and checked throughout.

Round 2

Reviewer 4 Report

The authors improved the quality of the article.

I recommend to publish this review.